# Investigation of Inkjet-Printed Masks for Fast and Easy Photolithographic NIL Masters Manufacturing

**DOI:** 10.3390/mi14081524

**Published:** 2023-07-29

**Authors:** Selina Burkert, Christian Eder, Andreas Heinrich

**Affiliations:** Center of Optical Technologies, Aalen University, 73430 Aalen, Germany; selina.burkert@hs-aalen.de (S.B.); christian.eder@hs-aalen.de (C.E.)

**Keywords:** nanoimprint lithography, photolithography, inkjet, photomask, master manufacturing

## Abstract

Modern optical systems often require small, optically effective structures that have to be manufactured both precisely and cost-effectively. One option to do this is using nanoimprint lithography (NIL), in which the optical structures are replicated as masters using a stamping process. It would also be advantageous to manufacture the master structures quickly and easily. A master manufacturing process based on a photolithographic image of an inkjet-printed mask is presented and investigated in this paper. An essential element is that a deliberate blurring of the printed structure edge of the mask is used in the photolithographic process. Combined with the use of a non-linear photoresist, this allows for improved edge geometries of the master structure. We discuss the inkjet-printed photomask, the custom photolithography system to prevent imaging of the printing dot roughness and the manufacturing processes of NIL polymer masks as well as their subsequent stamp imprinting. Finally, it was shown that stamp geometries with a width of 1.7 µm could be realised using inkjet-printed photomasks in the master manufacturing process. This methodology opens up the potential of fast and simple master manufacturing for the development and manufacturing of optical elements.

## 1. Introduction

In the field of nanofabrication, nanoimprint lithography (NIL) has recently become more popular [1,2,3,4]. NIL allows master structures to be replicated with high precision in a short time using a working stamp [5,6,7,8]. Thereby, features down to 10 nm can be replicated [9]. This makes NIL particularly interesting for the realization of optical [6,10], optoelectronic [11,12] and electronic devices [13].

The NIL process is based on master fabrication, stamp fabrication and replication, including the selection of suitable materials. A basic distinction is made between soft and hard NIL. For soft NIL, polymers such as polymethylmethacrylate or polydimethylsiloxane are used as the stamp material. In the hard NIL process, (semi)metallic stamps made from silicon or chrome are applied. Furthermore, a distinction is also made in terms of the replica’s curing. The polymer of the replica can be fixed with heat (hot embossing) as well as with ultraviolet (UV) light. A detailed description of the NIL technique can be found, e.g., in [1,14,15,16]. This article focuses on the master and subsequent stamp manufacturing for a UV–soft NIL process.

Master structures with sizes in the micro- and nanometre range are usually written into silicon (Si) or polymer substrates using focused ion beam (FIB) lithography, electron beam lithography (EBL), or photolithography [17].

On one hand, a minimal structure size of 3 nm to 200 nm can already be realised using the aforementioned high-end systems of FIB [18,19], EBL [20] and extreme ultra violet photolithography [21,22]. Thus, the clear advantage of these systems is their resolution in the nanometre range. On the other hand, the respective process duration and cost depend on the used technique, structure size and writing field, among other things. In general, the processes are slow and the costs are high [23]. For example, writing a field of 0.42 mm^2^ using an FIB system (available Carl Zeiss Auriga system, Ga^+^, 30 kV 4 nA milling current, 0.5 µm depth, silicon substrate) requires 16 h. This is quite often not acceptable for flexible low-cost master fabrication in the micrometre range as a higher resolution is not needed. 

A well-known technique that allows for a flexible low-cost master fabrication is inkjet printing. Inkjet printing is commonly used for the additive manufacturing of biological [24], optical [25] and electronic devices [26]. Etch masks have been printed directly onto substrates to create three-dimensional structures [27,28]. This technique has two major disadvantages. The resolution is directly related to the print quality on the substrate to be etched. In addition, direct printing is very time consuming compared with an exposure process with a photolithography system. It is more time efficient to use an inkjet-printed photomask several times for the quick exposure of a master structure.

In this article, we present the utilization of an inkjet-printed photomask and a custom photolithography system in the master manufacturing process to enable fast and precise master structures. We investigate the master manufacturing as well as the following NIL process with structures sizes in the single-digit micrometre range. A decisive advantage of an inkjet-printed photomask is that different masks can be printed quickly. For example, the mask for a comparable exposed field of 0.42 mm^2^ is printed in less than 1 min (Dimatrix Material Printer, 1.5 × 10^−1^ m/s at 10 kHz jetting frequency, 15 μm drop spacing). The total process time for preparation, exposure and development of the master structure is less than 10 min. In comparison with the mentioned FIB system, the presented photolithographic process with an inkjet-printed photomask is 96 times faster. However, a printed photomask has one critical disadvantage. The printed structure edges show relatively high edge roughness due to the individual printing dots of the inkjet process. This roughness is transferred to the master structure during exposure. In this article, we discuss a photolithography system for imaging inkjet-printed masks that eliminates this disadvantage. Deliberately reducing the resolution of the imaged mask structure prevents the edge roughness from being imaged into the photoresist. To improve the edge geometries and the resolution of the master structures, a non-linear photoresist is used that only develops above a certain intensity threshold. It is a known technique from photolithography to develop sharp structures based on blurred exposed structures, as explained in [29]. 

We discuss below the inkjet-printed photomask, the custom photolithography system to prevent the imaging of printed point roughness and the manufacturing processes of NIL polymer masters as well as their subsequent stamp imprinting.

## 2. Inkjet-Printed Photomask

Since the design of the photolithography system is based on the edge roughness of the inkjet-printed photomask, the printing system and the mask were analysed first. The aim here was to evaluate the minimum structure width and the edge roughness on the inkjet-printed photomask. The photomask was printed using an inkjet printer (Dimatix Material Printer 2850, FUJIFILM, Tokyo, Japan), black ink (Dimatix test ink, FUJIFILM) and a high-resolution film (OptiSytle, 135 µm thickness, Papyrus, Berkeley, CA, USA). The precision of the inkjet printer used depends, among other things, on the ink, the nozzle array of its printhead and the selected print line spacing. The print line spacing is the distance between two horizontal printed lines. It has a dominant impact on the print quality as an adjustable parameter. The applied ink volume for a printed area is set via the print line spacing. If the print line spacing is too small, too much ink is applied to an area. The individual drops crosslink to form large drops. The centroids of the drops are further apart and drop diameters have increased. This leads to an increase in the roughness of the printed edge. If the print line spacing is too large, the ink on the printed area is not sufficient for crosslinking. Instead of an area, individual drop lines are printed. Due to the lack of crosslinking, the roughness is also increased in this case. The print edge roughness is quantified through the simple standard deviation along a print edge. A minimum print edge roughness of ±2 µm was achieved using a drop spacing of 15 µm. This yields a print line spacing of 15 µm for the photomask printing.

To evaluate the minimum structure width and the edge roughness as a function of the structure size, a periodic test pattern, which is shown in Figure 1, was designed. Both positive and negative (inverted) printed structures were used. The bar width w depends on the selected print line spacing of 15 µm. This results in bar widths w of 15 μm to 135 μm in 15 µm steps.

Each bar group was captured by a white light interferometer (WLI) (NewView8300, Zygo, Middlefield, CT, USA) and subsequently evaluated regarding its bar width and edge roughness using MATLAB. Based on the negative 135 µm structures, Figure 2 illustrates how these parameters were determined. The mean structure width corresponds to the average structure width w_i_ of all bars, which was measured over 80% of the total bar length l. As shown in Figure 2, the imaged edges were detected using a ‘Canny’ operator (1.) to evaluate the printed edge roughness. Misinterpretations of the edge detection that were not part of the edge line were suppressed with a binary mask ((2.), e.g., top left corner). The remaining edge line (3.) was used to calculate the standard deviation of the edge roughness.

The results of the analysis of the printed photomasks are summarised in Figure 3. Figure 3a,b show the mean bar width and the mean print edge roughness along the print edges. Since the printed structures are only visible starting from an expected bar width of 45 µm, the analysis was restricted to a range of 45 µm to 135 µm. Based on Figure 3a, a systematic offset obviously exists with respect to the ideal bar width of +27.5 (±1.2) µm for positive and −25.7 (±0.9) µm for negative structures. This is due to the fact that the position of the ideal print edge is related to the centre of a single print dot. Since drops are deposited during the inkjet process, this edge shifts outwards by half the drop diameter for positive structures and inwards for negative structures. This results in a deviation from the ideal structure width. This may be taken into account in the mask design. The smallest possible structure width is 18.1 ± 2 µm for negative structures. Figure 3b shows that the standard deviation of the edge roughness spreads evenly and is independent of the structure size and type (positive, negative). The mean edge roughness is 2 µm.

## 3. Design and Parameters of the Lithography System

The realised photolithography system is shown in Figure 4. It consists of an illumination unit, a deliberate pre-reduction below a defined resolution limit (see Section 3.2), which results in a blurred image, and a main reduction. The illumination beam path (red) and the imaging beam path (blue) are drawn into the photolithography system shown. The corresponding image planes and the intensity profiles are illustrated to the right of the system. A camera that can be used to focus the photolithography system is attached.

### 3.1. Illumination Beam Path

To image the light source into the field stop of the macro lens, the illumination beam is expanded using a Kepler telescope (f = 35 mm D = 24.5 mm N-BK7 or f = 150 mm D = 24.5 mm N-BK7, Thorlabs, Newton, NJ, USA). Using the Kepler telescope meant that vignetting was avoided, whereas the usable area in the substrate plane and the illumination homogeneity were doubled. The system’s illumination power is controlled by an LED controller via the pulse width modulation signal of an ADRUINO nano and determined to 165 mW/cm^2^ in the substrate plane using a power meter (S142C, Thorlabs). The required exposure time may be calculated based on the specified exposure dose for each polymer.

### 3.2. Imaging Beam Path

The blurring and pre-reduction unit consists of a macro lens (Macro Video Zoom Lens 18-108 F/2.5, Optem). The most optimal lens setting of field stop, working distance and zoom was determined taking into account the resolution (MTF analysis, resolution limit at 0.2 of contrast), field of view (magnification analysis) and radial distortion (point test pattern analysis). The maximum resolution was determined by means of the print system analysis. Statistically, the determined edge roughness of ±2 µm in the form of the simple standard deviation covers only 34.1% of the edge roughness. To ensure that the total edge roughness is eliminated by the system design, the total edge roughness is assumed to be six times the standard deviation of 12 µm. The total edge roughness of 12 µm corresponds to a spatial frequency of 83.3 lp/mm. A resolution limit of 83.3 lp/mm was set for the blurring and pre-reduction unit so that the photolithography system no longer resolves the total edge roughness. Thus, the blurring and pre-reduction unit intentionally reproduces the structure of the mask on the intermediate image plane in a blurred and slightly reduced form, as is also shown in Figure 5.

The main reduction unit comprises a tube lens (f = 175 D = 24.5 mm mm N-BK7, Thorlabs) and a microscope lens (HC PL FLU-OTAR 20x/0.50, Leica, Wetzlar, Germany), which images the intermediate image onto the substrate via the tube lens and the beam splitter. A positive photoresist with a non-linear response function (binary resist, ma-P 1240, MicroResist, Berlin, Germany) is used as the substrate. Due to the non-linear response function, only areas in which a certain exposure intensity has been exceeded are developed. Thus, steep slopes occur in the non-linear polymer caused by a smoothed intensity distribution, as shown by the intensity distribution in Figure 5 (intensity distribution in the resist plane).

The substrate is captured by a camera (mvBlueFox-1100G, Matrix Vision, lens MLH-10x, Computar). The camera system allows the substrate to be positioned in the focus of the lithography system with a three-axis stage (x- and y-stage M-531.PD, z-stage M-501.DG, Physik Instrumente, Karlsruhe, Germany).

### 3.3. Parameterising the Photolithography System

The subsystems were examined with regard to their resolutions and magnifications for system qualification. This was carried out using a resolution test pattern (USAF chart). The resolution test pattern used is shown in Figure 5. It features periodic test groups that map spatial frequencies from 1 lp/mm to 228 lp/mm. When investigating the resolution, the resolution test pattern was placed on the mask plane and an MTF analysis was performed based on the imaging of the system plane of interest, such as the substrate plane (see Figure 6). The magnification factor was determined using the ratio between the bar width in the respective imaging plane and the resolution test pattern. 

The determined resolution and the corresponding magnification factor are summarised in Table 1. First, the resolution and magnification of the substrate structure was measured on the camera sensor used to analyse the system. This was performed by placing the resolution test pattern on the substrate plane. The camera system resolution was found to be greater than the test pattern’s maximum spatial frequency of 228 lp/mm, resulting in a magnification factor of 6. 

The imaging of the resolution test pattern from the mask plane to the substrate plane was investigated in terms of resolution and magnification. The measured camera magnification was taken as a basis here. A system resolution of 29 lp/mm was measured using the MTF analysis of the substrate plane image shown in Figure 6. This is over seven times smaller than the camera system’s resolution. Thus, the camera system is suitable for analysis and system alignment. From the mask plane to the substrate plane, the lithography system has a magnification factor of 0.03. Due to the mount of the second Kepler lens, the exposed mask diameter is limited to 24.3 mm. The spot diameter in the substrate plane is 0.73 mm, which corresponds to an exposed area of 0.43 mm^2^. 

The resolution and magnification of the blurring and pre-reduction unit were examined in the intermediate image plane. At a magnification factor of 0.5, the resolution is 55.2 lp/mm. This results in a calculated magnification factor for the main reduction unit of 0.06.

Finally, the interaction between the photolithography system and the non-linear resist was investigated, as the threshold value of the resist also determines the resolution of the master structure. The resolution and magnification were analysed using a developed USAF test chart in the non-linear resist. The result was a resolution of 30.6 lp/mm and a magnification factor of 0.03 (edge lengths of the mask and exposed resist structure were 556 µm and 16.62 µm, respectively).

The results in Table 1 show that the resolution in the polymer at 30.6 lp/mm is 5% higher than that of the lithography system at 29 lp/mm. This improvement can presumably be attributed to the non-linearity of the photoresist. This causes an improvement in the contrast during exposure, allowing higher spatial frequencies to be transmitted.

## 4. NIL Master Manufacturing

A polymer master for a NIL process was produced using the photolithography system and the inkjet photomask shown in Figure 1. Negative structures with widths of 18.1 ± 2 µm to 108.9 ± 2 µm in 15 µm steps were used. The polymer master was realised in three basic steps (see Figure 7): substrate preparation (a), exposure (b) and development (c). The associated process parameters are summarised in Table 2. First, a 4 borosilicate wafer was baked out for dehydration and a hexamethyldisilazane adhesion promoter (HMDS, MicroResist) was spun. The resist (ma-P 1240, MicroResist) was then spun, followed by a short pre-bake (a). In the next step, the polymer structure was exposed in the photolithography system through the inkjet photomask (b). In the last step, the exposed structure was removed in the developer bath and cleaned with distilled water (c). 

The developed polymer structures were measured using a WLI (20×, NA 0.4, lat. res. 0.4067 µm/px, vert. res. 0.1 nm, Zygo) and evaluated in MATLAB with respect to structure height and width. To illustrate the image transfer and the evaluation, this is shown as an example in Figure 8 for a mask structure width of 108.9 ± 2 µm. Based on the topographic images of the WLI, a polymer layer height of 4.13 ± 0.03 µm was determined. At the height of the full width half maximum (FWHM), the structure widths were determined at 2.06 µm (red contour). 

The structure was found to be completely ablated down to the substrate starting from a structure width of 2.1 ± 0.1 µm. With a polymer layer thickness of 4.13 ± 0.03 µm, this results in a maximum aspect ratio of 0.51. Smaller structures could only be partially developed. Taking into account the maximum resolution in the polymer of 30.6 lp/mm from Section 3.3 and the resulting theoretical structure width of 0.49 µm, a polymer height of 1 µm is recommended for a maximum aspect ratio of 0.51. 

The investigations into the developed structure widths showed that the magnification factor of the overall system (photolithography system + resist) depends on the mask structure size. This behaviour is shown in Figure 9 based on the magnification factor for the exposed structures through the inkjet-printed photomask. The smaller the mask structure, the greater the relative influence the blurring will exert on the structure width. In small mask structure sizes, this causes the magnification factor to decrease. From a structure width of 120 µm, a maximum magnification factor of 0.03 is reached and the blurring has a negligible influence on the structure size. The magnification m as a function of the structure size w may be approximated with the function m = 3.4 w^−1.4^ + 0.03 and taken into account in the mask design.

## 5. NIL Stamp Manufacturing

The polymer master was used to make a NIL stamp to further replicate the master structure. The NIL stamp manufacturing process is shown in Figure 10 and consists of three basic steps: substrate preparation (a), exposure (b) and separation of the stamp and master (c). The detailed process parameters are provided in Table 3. The 4” borosilicate wafer is baked out for dehydration. After cooling, the borosilicate wafer is coated with an adhesion promoter (PrimerK, EV Group, St. Florian am Inn, Austria) and briefly pre-baked (a). Next, the stamp material (UV/AS1, EV Group) is applied to the master and topped with a borosilicate wafer. The stamp material is cured in a mask aligner (EVG 620) (b). Finally, the wafer is separated from the master using a blade. Any remaining master material on the stamp is chemically dissolved in a PGMEA bath and rinsed off (c).

The stamp structures were measured using the WLI and an environmental scanning electron microscope (ESEM). The measurement data were evaluated with respect to their bar width using MATLAB and a raw data evaluation software (Gwyddion). The framed mask structures (w = 18.1 to w = 49.7) from Figure 11a are shown in Figure 11b in top view (ESEM image at 45°). The smallest realised NIL stamp structure (w = 18.1 µm) is shown in magnified format in c) (ESEM image at 90°). This is the smallest printable mask structure.

Since the master structure is evaluated regarding its FWHM value, the FWHM value was measured indirectly. Assuming that the bottom and the plateaus are linearly connected, the average of the base and plateau width (see measuring lines) was measured. The calculated structure width is 1.73 ± 0.08 µm. When comparing a structure width of 18.1 ± 2 µm on the inkjet photomask with the realised stamp structure width of 1.73 ± 0.08 µm, the structure was reduced by a factor of 0.09.

The manufactured NIL stamps have a layer thickness of 3.93 ± 0.03 µm. The maximum realised aspect ratio is therefore 0.54. Compared with the aspect ratio of the smallest fully developed master structure of 0.51, it is clear that the aspect ratio of the stamp is 0.03 (5.6%) larger. This is due to the shrinkage behaviour of the stamp material and can already be taken into account in the master design by means of an iterative process. The use of an inkjet-printed photomask in master manufacturing offers a quick and easy approach here.

## 6. Conclusions

A customised photolithography system for the application of inkjet-printed masks in the photolithographic master manufacturing process was presented in this article. The photolithography system consists of an illumination unit, a blurring and pre-reduction unit (macro lens) and one of the main reduction units (microscope lens). It intentionally resolves below the resolution limit of 83.3 lp/mm derived from the print edge roughness so that the edge roughness of the inkjet-printed mask structure is not transferred to the NIL master. A non-linear photopolymer is used as the master material, which provides contrast enhancement and allows for the realisation of small structures with steep edges. Using a nonlinear master polymer such as this one, the resolution of the photolithography system was improved by 5% from 29.6 lp/mm to 30.6 lp/mm. This theoretically corresponds to a minimum structure width of 0.49 µm in the NIL master. Furthermore, it was shown that, due to the blurring, the magnification factor of the overall system (photolithography system + resist) depends on the mask structure size. A stamp with medium soft UV NIL was moulded from the developed polymer master. Its minimum structure width was 1.73 ± 0.08 µm.

## Figures and Tables

**Figure 1 micromachines-14-01524-f001:**
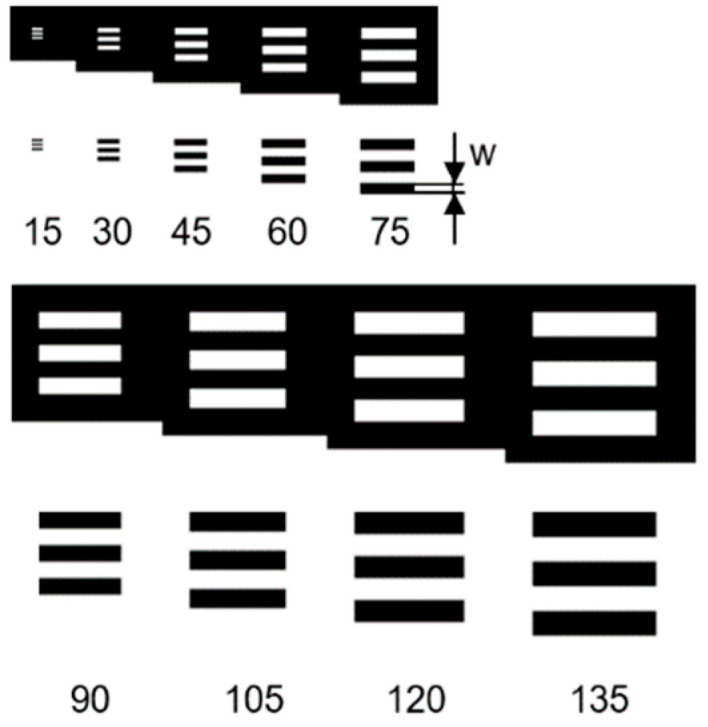
There are positive and negative bars on the digital mask layout. Bar widths w from 15 µm up to 135 µm in 15 µm steps were tested.

**Figure 2 micromachines-14-01524-f002:**
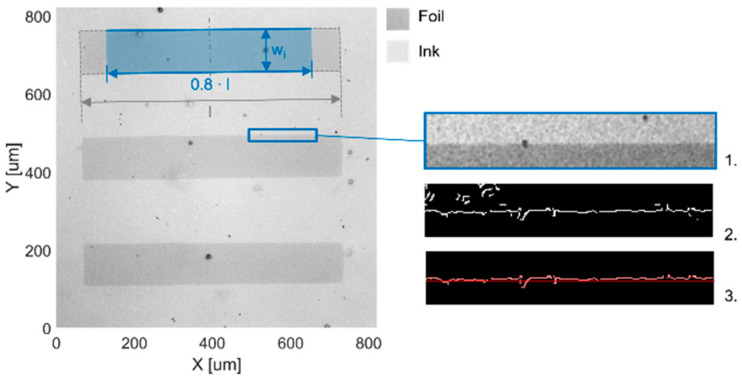
The WLI measurement of 135 µm printed negative structures. The measured bar width is calculated based on the average bar width w_i_ of the three bars along 80% of the total bar length l. The edge roughness is defined by the standard deviation of the printed edge as demonstrated on the right side.

**Figure 3 micromachines-14-01524-f003:**
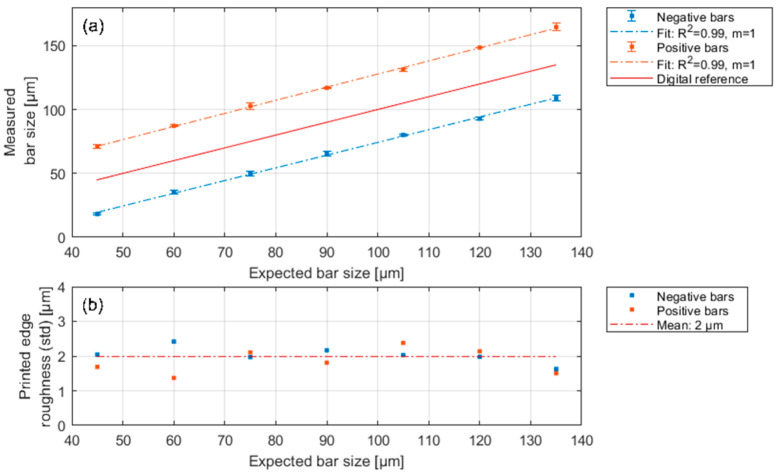
(**a**) Measured bar size versus expected bar size. The positive bars are 27.5 µm wider and the negative structures are 25.7 µm narrower than expected due to the ink dots’ diameter. The mask design can be improved by taking the constant offset into account. (**b**) Dependency of the mean standard deviation on the ink dot roughness. The roughness along the structures is 2 µm.

**Figure 4 micromachines-14-01524-f004:**
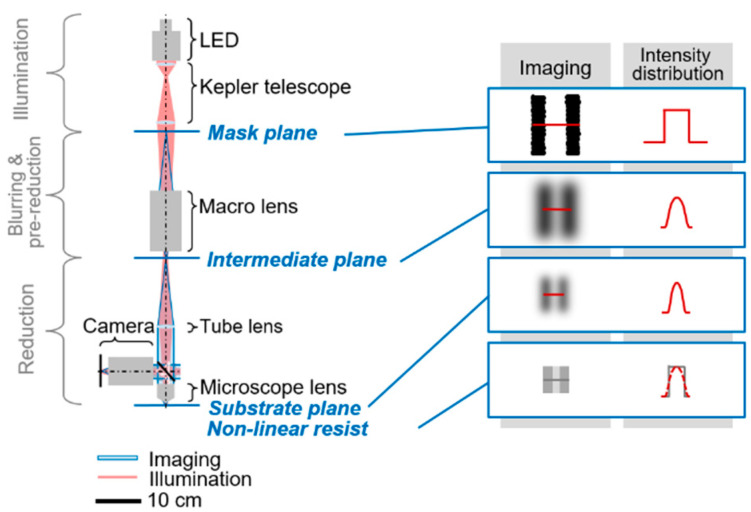
The photolithography system is based on an illumination unit, a blurring and pre-reduction unit (macro lens) and a high-resolution reduction unit (microscope lens). The respective imaging and the intensity distribution of the image planes are shown on the right side.

**Figure 5 micromachines-14-01524-f005:**
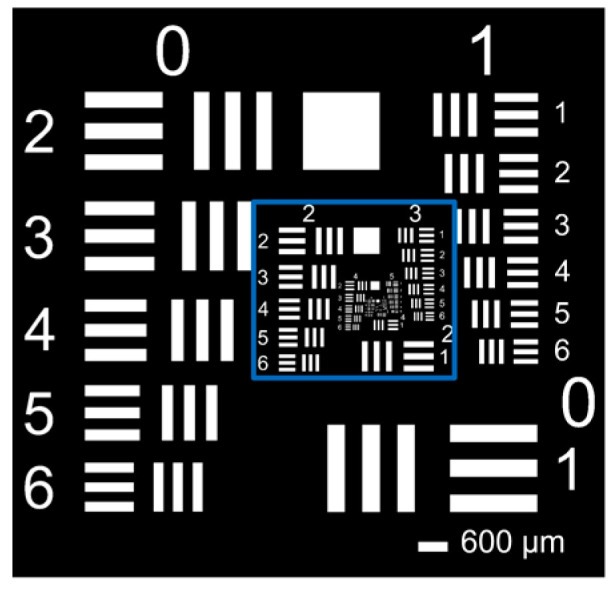
The applied resolution test chart (USAF chart) covers a range from 1 lp/mm (Group 0 Element 1) up to 228 lp/mm (Group 7 Element 6).

**Figure 6 micromachines-14-01524-f006:**
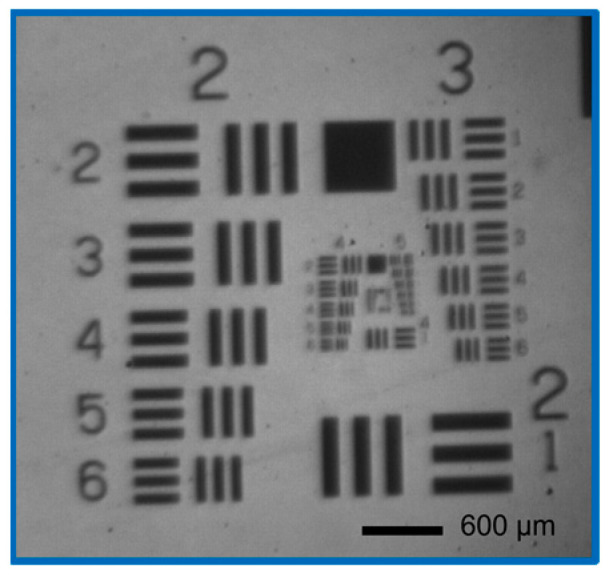
Transferred image of the resolution test chart in the substrate plane.

**Figure 7 micromachines-14-01524-f007:**
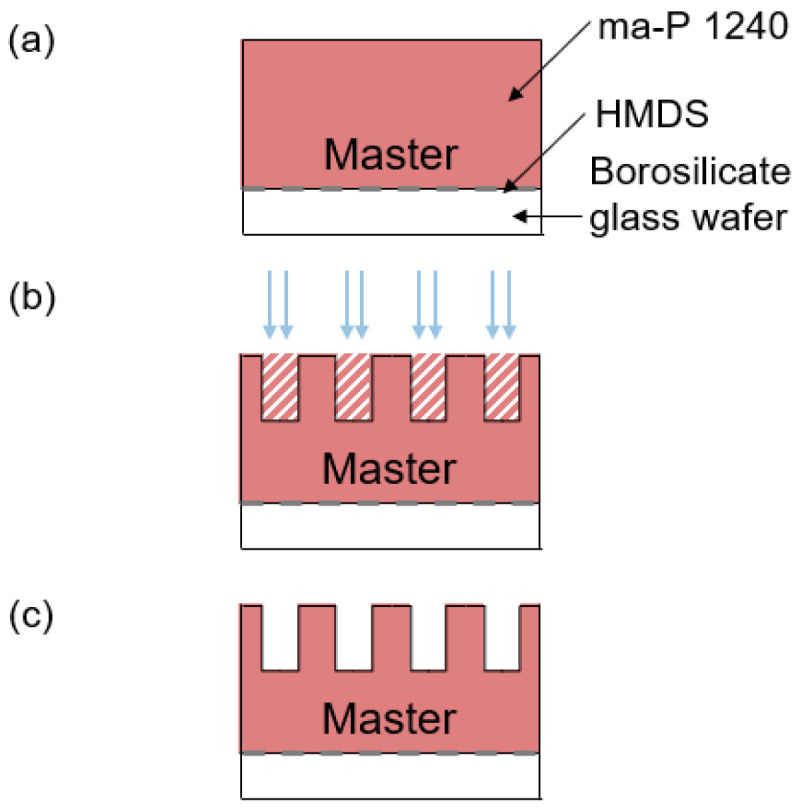
(**a**) The master manufacturing was based on three steps. First, the master substrate was spin-coated with HMDS primer and the photoresist ma-P 1240. (**b**,**c**) Then, the structure was exposed by the individual lithography system to UV light and developed.

**Figure 8 micromachines-14-01524-f008:**
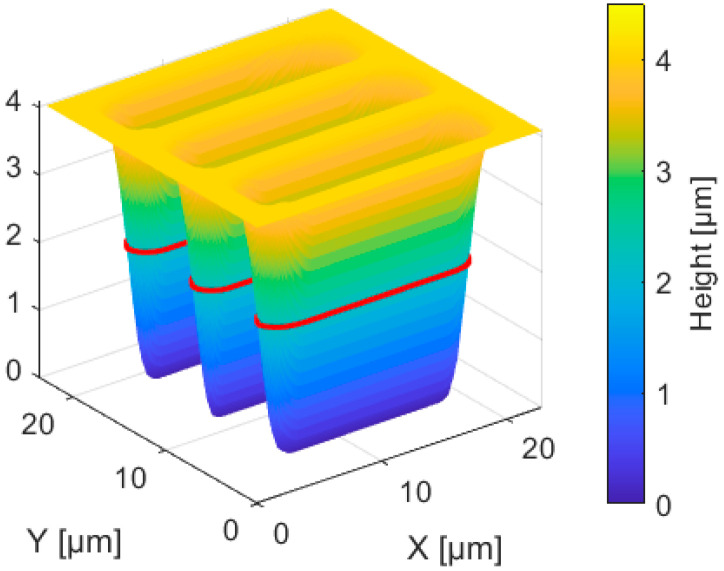
Measured polymer master structure of the 108.9 ± 2 µm mask bars resulting in a structure width of 3.4 µm (FWHM, red line).

**Figure 9 micromachines-14-01524-f009:**
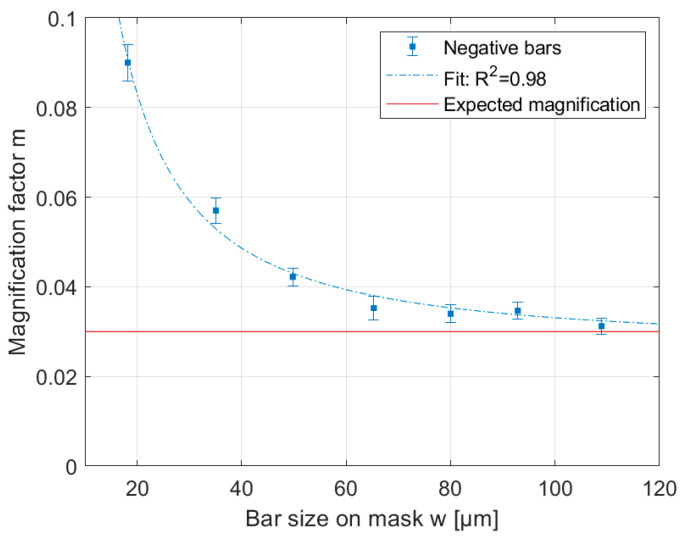
The magnification factor m depends on the mask structure’s size w and can be approximated by m = 3.4 w^−1.4^ + 0.03.

**Figure 10 micromachines-14-01524-f010:**
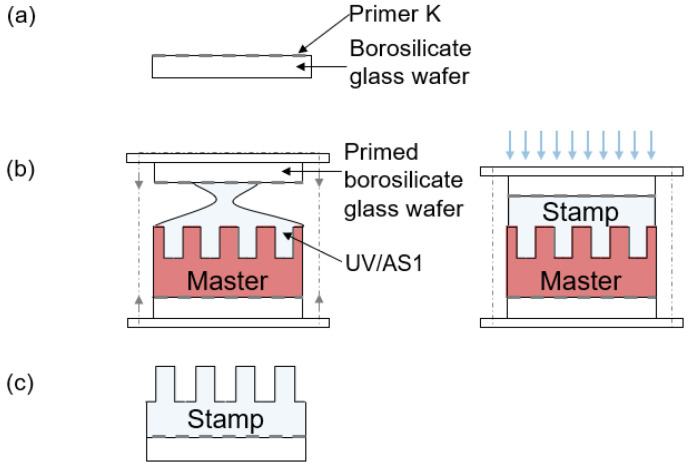
(**a**) The stamp manufacturing process is conducted in three steps. First, the stamp is coated with an adhesion promoter (Primer K). (**b**) Then, the stamp material is placed between the coated stamp substrate and the master using a two-part chuck. The stamp material is cured by UV light. (**c**) Finally, the master material is mechanically and chemically removed from the stamp.

**Figure 11 micromachines-14-01524-f011:**
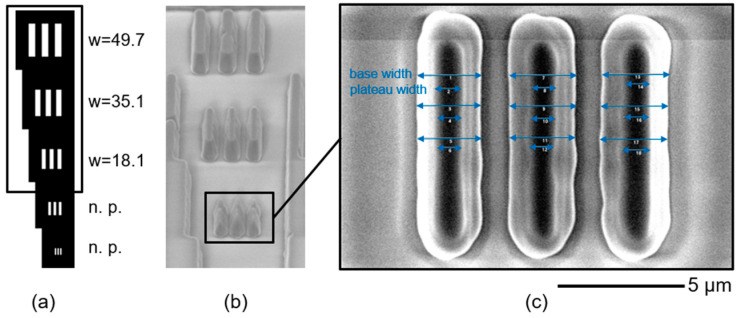
(**a**) Schematic drawing of the mask of the mask. (**b**) Captured ESEM image. (**c**) Smallest transferred structure (which is equal to the smallest printed negative mask structure). Smaller structures are not printable (n.p.).

**Table 1 micromachines-14-01524-t001:** The resolution in different system planes and in the polymer and the corresponding magnification factors.

System Plane	Resolution (lp/mm)	Magnification References	Magnification
Camera sensor	>228	Substrate plane—camera sensor	6.1
Intermed. image plane	55.2	Mask—intermediate image plane	0.5
Substrate plane	29	Mask—substrate plane	0.03
In the resist	30.6	Mask—polymer	0.03

**Table 2 micromachines-14-01524-t002:** NIL master manufacturing process parameters.

Process	Parameter	Setting
Wafer bake	Temperature (°C), Time (s)	200, 1800
Cooling	Time (s)	approx. 1800
Cool down by spinning	Spin speed (rpm), Time (s)	500, 45
	Spin speed (rpm), Time (s)	4000, 2
Spin Coating HMDS	Volume (mL)	approx. 0.2
	Spin speed (rpm), Time (s)	500, 45
	Spin speed (rpm), Time (s)	4000, 2
Spin Coating ma-P 1240	Volume (mL)	5
	Spin speed (rpm), Time (s)	3000, 30
Prebake	Temperature (◦C), Time (s)	100, 300
Exposure	100% LED power (mW/cm^2^), Time (s)	165, 0.7
Development with mr-D 331	Time (s)	47
Rinse with distilled water and dry by spinning	Spin speed (rpm), Time (s)	500, 45
	Spin speed (rpm), Time (s)	4000, 2

**Table 3 micromachines-14-01524-t003:** Parameters of the NIL stamp manufacturing process.

Process	Parameter	Setting
Wafer bake	Temperature (°C), Time (s)	200, 1800
Cooling	Time (s)	approx. 1800
Cool down by spinning	Spin speed (rpm), Time (s)	500, 45
	Spin speed (rpm), Time (s)	4000, 2
Spin Coating Primer K	Volume (mL)	approx. 0.2
	Spin speed (rpm), Time (s)	500, 45
	Spin speed (rpm), Time (s)	4000, 2
Bake	Temperature (°C), Time (s)	120, 120
Applying the stamp material on the master	Volume (mL)	0.6
Expose with min. 2000 mJ/cm^2^	EVG 620 100% LED power (mW/cm^2^), Time (s)	20, 600
Ma-P 1240 dissolving by PGMEA	Time (s)	1800
Rinse with distilled water and dray by spinning	Spin speed (rpm), Time (s)	500, 45
	Spin speed (rpm), Time (s)	4000, 2

## Data Availability

The measurement data and further details on the experiment can be requested via the mentioned contact data.

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
