# Peer review of "Investigation of Inkjet-Printed Masks for Fast and Easy Photolithographic NIL Masters Manufacturing"

_micromachines, 2023, doi:10.3390/mi14081524_

Round 1

Reviewer 1 Report

The authors demonstrated the use of inkjet technique for the fabrication of the master of the NIL. The use a strategy to avoid transferring the print edge roughness to the NIL master. Overall, this work is merit for the journal.

Following is some issues need to be addressed before considering acceptation.

1, as has wrote by the authors in the introduction, there are many well-established techniques, such as FIB, EBL and Photolithography for fabrication the master structures in micro and nanometer range, what is the advantage and disadvantage of inkjet technique compared with those well-established techniques? A comprehensive analysis considering the resolution, cost and the efficiency would be preferred.

2, Inkjet printing is a very wildly used technique for printing various inks on various substrate, including printing masks. However, I did not see reference about using inkjet for fabricating masks.

3, the authors said that the optimal print line spacing should be selected otherwise the print edge roughness will be larger. I suggest the authors provided a more detailed explanation of this phenomenon. Does the arrangement of the orifices on the nozzle plate has any relation with this phenomenon?

4, how thick is the printed film? Does the thickness of the printed film matters? Beside the edge roughness, more description about the printed patterns need to be provided.

Reviewer 2 Report

This article introduced a customised photolithography system for the application of inkjet-printed masks in the photolithographic master manufacturing process. The author investigated the mechanism of the non-linear photopolymer on the resolution of the photolithography system, the custom photolithography system to prevent imaging of the printing dot roughness, and the manufacturing processes of nanoimprint lithography polymer masks as well as their subsequent stamp imprinting. The author's ideas have some novelty, and the results explained are important for researchers working in the specific field. However, some content of the paper still needs additional modification and improvement before being prepared for publication. Please see the comments below:

1. The author investigated the interaction of the photolithography system and the non-linear resist based on the decisive relationship between the threshold value of the resist and the resolution of the master structure. This decisive relationship should be briefly described.

2. For the results in Table 1, the improvement of the resolution in the polymer relative to that of the lithography system is possibly attributed to the non-linearity of the photoresist. This hypothetical relationship should be introduced in the introduction.

3. In Figure 8, the magnification m as a function of the structure size w is approximated with the equation m = 111.4 w-2.3 + 0.03. However, there are only six data supporting the equation. More data should be displayed to improve the reliability of this equation.

4. For more clarity, Figure 5(b) should be tagged extra.

Reviewer 3 Report

A minor editing of language is required

Round 2

Reviewer 1 Report

The revisions made by the authors are satisfying now.

Author Response

Thank you for your support.

Reviewer 3 Report

The authors are supposed to add the mentioned references in the article. There is no point of reviewing literature and checking the robustness of the article by a reviewer if the authors are not ready to add relevant researches in the articles.

Also, the authors precisely try to keep the introduction as short as possible; the reason for which is not understood. Writing a line about a technique is not enough. Or providing one example of a research in that field is not enough.
